# Characterization of Growth-Promoting Activities of Consortia of Chlorpyrifos Mineralizing Endophytic Bacteria Naturally Harboring in Rice Plants—A Potential Bio-Stimulant to Develop a Safe and Sustainable Agriculture

**DOI:** 10.3390/microorganisms11071821

**Published:** 2023-07-16

**Authors:** Md. Yeasin Prodhan, Md. Bokhtiar Rahman, Aminur Rahman, Md. Ahedul Akbor, Sibdas Ghosh, Mst. Nur-E-Nazmun Nahar, Md. Shamsuzzoha, Kye Man Cho, Md. Azizul Haque

**Affiliations:** 1Department of Biochemistry and Molecular Biology, Hajee Mohammad Danesh Science and Technology University, Dinajpur 5200, Bangladeshbokhtiarbdj@gmail.com (M.B.R.); nnnbhstu@gmail.com (M.N.-E.-N.N.); tahminasimo148@gmail.com (S.); 2Department of Biomedical Sciences, College of Clinical Pharmacy, King Faisal University, Al-Ahsa 31982, Saudi Arabia; 3Institute of National Analytical Research and Services (INARS), Bangladesh Council of Scientific and Industrial Research (BCSIR), Dhaka 1205, Bangladesh; akborbcsir@yahoo.com; 4Department of Biological Sciences, College of Arts and Sciences, Carlow University, 3333 Fifth Avenue, Pittsburgh, PA 15213, USA; sghosh@carlow.edu; 5Department of Chemistry, Hajee Mohammad Danesh Science and Technology University, Dinajpur 5200, Bangladesh; ms_zoha2006@yahoo.com; 6Department of Green Bio Science and Agri-Food Bio Convergence Institute, Gyeongsang National University, Jinju 52725, Republic of Korea; kmcho@gnu.ac.kr

**Keywords:** pesticide-degrading endophyte, growth promotion, MDR bacterial inhibition, synthetic consortia, GC–MS/MS analysis, rice plant, yields enhancement

## Abstract

Eighteen pesticide-degrading endophytic bacteria were isolated from the roots, stems, and leaves of healthy rice plants and identified through 16S rRNA gene sequencing. Furthermore, biochemical properties, including enzyme production, dye degradation, anti-bacterial activities, plant-growth-promoting traits, including N-fixation, P-solubilization, auxin production, and ACC-deaminase activities of these naturally occurring endophytic bacteria along with their four consortia, were characterized. *Enterobacter cloacae* HSTU-ABk39 and *Enterobacter* sp. HSTU-ABk36 displayed inhibition zones of 41.5 ± 1.5 mm, and 29 ± 09 mm against multidrug-resistant human pathogenic bacteria *Staphylococcus aureus* and *Staphylococcus epidermidis*, respectively. FT-IR analysis revealed that all eighteen isolates were able to degrade chlorpyrifos *pesticide.* Our study confirms that pesticide-degrading endophytic bacteria from rice plants play a key role in enhancing plant growth. Notably, rice plants grown in pots containing reduced urea (30%) mixed with either endophytic bacterial consortium-1, consortium-2, consortium-3, or consortia-4 demonstrated an increase of 17.3%, 38.6%, 18.2%, and 39.1% yields, respectively, compared to the control plants grown in pots containing 100% fertilizer. GC–MS/MS analysis confirmed that consortia treatment caused the degradation of chlorpyrifos into different non-toxic metabolites, including 2-Hydroxy-3,5,6 trichloropyridine, Diethyl methane phosphonate, Phorate sulfoxide, and Carbonochloridic. Thus, these isolates could be deployed as bio-stimulants to improve crop production by creating a sustainable biological system.

## 1. Introduction

Agriculture remains one of the most important economic sectors in Bangladesh and plays a crucial role in the rural economy of the country. Thus, the performance of this sector is critical to irradicating poverty. Excessive use of pesticides discharging into surrounding water ultimately impregnates populations through direct and indirect contact, leading to fatal non-communicable human diseases [1]. Furthermore, the excessive importation of fertilizers and pesticides contributes to the interruption of reserve dollar in the national economy and an increase in the prevalence of non-communicable diseases. All in all, this interferes with the sustainable development goals of Bangladesh. Continuing to achieve a steady increase in food production along with human health requires favorable weather conditions and efficient use of fertilizers along with measured application of biological control systems for pests, insects, and fungi. One such example includes the use of growth-promoting endophytic bacteria with capabilities of degrading pesticides.

Endophytic bacteria ubiquitously colonize the internal tissues of plants and are found in nearly every plant worldwide, which can promote the growth of plants through increased germination rates, biomass, leaf area, chlorophyll content, nitrogen content, protein content, hydraulic activity, roots and shoot length, yield and tolerance to abiotic stresses, including but not limited to drought, flood, and salinity [2,3]. Furthermore, endophytic bacteria have been demonstrated to promote plant growth directly through biological nitrogen fixation, phytohormone production, phosphate solubilization, inhibition of ethylene biosynthesis under both biotic or abiotic stresses (induced systemic tolerance), or indirectly through inducing resistance to the pathogen [4]. Some of these endophytes are reported to have generated adaptation against fungicides, herbicides, insecticides, and pesticides [5] by their abilities to degrade and/or metabolize these organic-based synthetic control agents [6,7]. The biochemical mechanisms involving degradation of organophosphate pesticides include adsorption, hydrolysis of P–O alkyl and aryl bonds, photodegradation, and enzymatic mineralization. Microbe-specific enzymes, including esterase, organophosphorus hydrolase, amidohydrolase, carboxylesterase, phosphotriesterase [6,8], diisopropyl fluorophosphatase, parathion hydrolase, and paraoxonase [9,10], have been demonstrated to be involved in the degradation of insecticides.

Rice is the staple food grain for more than 3.5 billion people around the world, particularly in Asia, Latin America, and parts of Africa, and serves as an important source of fiber, energy, minerals, vitamins, bioactive compounds, among other biomolecules [11,12]. Due to high year-round demands, farmers extensively adopt high-yielding varieties for increased production. However, the commercial farming of this vital crop is under immense threat from pests, insects, and diseases along with both biotic and abiotic stresses, leading to annual loss of yield up to 50% globally [13]. According to the Bangladesh Bureau of Statistics (2021), 54,500 metric tons of pesticide was applied in 2020, of which 20,896 metric tons was insecticides, to combat pests and diseases. In 2022, the Government of Bangladesh subsidized 300,000 million takas (over US $3M) to import additional fertilizers, a severe threat to the national economy. Therefore, controlling pests and pathogens in rice cultivation with potential consortia of pesticide-degrading endophytic bacteria might be a bio-solution ensuring enhanced yield along with the security of food and human health.

The present study aims to isolate and identify naturally occurring pesticide-mineralizing endophytic bacteria in rice plants with potential growth-promoting activities. A consortium of the isolates is identified to be used as a biofertilizer to increase the yield of rice with the following five objectives: (i) isolate and identify pesticide-mineralizing endophytic bacteria from the roots, shoots, and leaves of BRRI-28 and Kalijera rice plants from different fields, (ii) select the isolates showing accelerated plant-growth-promoting traits using biochemical analysis, (iii) determine the pesticide-mineralizing activities of the isolates in vitro using FT-IR and GC–MS, (iv) identify the inhibitory activities of the isolates against multidrug-resistant human pathogenic bacteria, and (v) conduct pot experiments under field conditions to determine the impact of consortia comprising the isolates on rice yields.

## 2. Materials and Methods

### 2.1. Sample Collection and Processing

Varieties of rice plants, *Kalijeera* and BRRI-28, were collected from two different paddy fields near Basher Hat, Dinajpur, Bangladesh. These locations were exposed frequently to organophosphate pesticides chlorpyrifos and diazinon for several years prior to our collections of the rice plants in 2020. These two locations were strategically selected to collect naturally occurring endophytic bacterial strains resistant to chlorpyrifos or capable of mineralizing chlorpyrifos. Two months after transplantation, healthy rice plants exposed to chlorpyrifos pesticide at least twice were selected, carefully cut into small pieces, and washed with tap water to remove soil and dust. Tissue samples were then surface sterilized with 75% ethanol for 3 min, followed by shaking in 1.2% (*w*/*v*) sodium hypochlorite (NaOCl) solution for 20 min. Samples were then washed thrice with sterile distilled water by shaking for 20 min each time. Only samples that demonstrated no infections, as described in ref. [14], were used for further experiments. Juices were extracted from each sterilized root, shoot, and leaf tissue separately using mortar and pestle and collected into sterilized test tubes for further analyses.

### 2.2. Screening for Pesticide-Degrading Bacteria

Endophytic bacteria were screened as previously described by Gyaneshwar et al. (2001) [13]. Briefly, the aliquot of juices from specific plant tissues was centrifuged at 1300 rpm at room temperature under aseptic conditions for 10 min. The supernatants were serially diluted up to 10^−5^, and each dilution was transferred to a pesticide-containing liquid medium in a conical flask and incubated at 37 °C with 130 rpm for 4 days. Then, the samples were cultured into the pesticide-containing medium as described [15]. The pesticide-degrading bacterial isolates were selected with the streak plate method containing 1 gm/100 mL of chlorpyrifos [7,15,16]. The isolates were selected based on distinct colony morphology and growth criteria, and this procedure was repeated several times until the pure colonies were achieved. Finally, the isolates of pure colonies were grown into tryptic soy broth liquid medium at 37 °C for 24 h and stored at 4 °C in the short term and at −20 °C in 50% (*w*/*v*) glycerol until further use.

### 2.3. Molecular Characterization and Phylogeny of Endophytic Bacteria

The genomic DNA of bacterial isolates was extracted as described by Haque et al. (2015) [14]. The amplification of 16S rRNA gene was performed using forward primer 27F 5′-AGA GTT TGA TCM TGG CTC AG-3′ and reverse primer 1492R 5′-GGT TAC CTT GTT ACG ACT T-3′ [15]. The primers, template DNA, and Taq DNA polymerase were added into the master mix right before loading the sample. After PCR reaction, the amplification was visualized by gel electrophoresis. After purification of the amplified 16S rRNA gene, the concentration was measured, diluted into 5 ng/µL, mixed with a ready reaction premix, and run for PCR sequences using genetic analyzer 3130 (Applied Biosystem, CL, Beverly, MA, USA). The resulting sequences were analyzed by BLASTn, submitted to NCBI, and the accession numbers were registered. The 16S rRNA sequences were compared to different 16S rRNA genes of other bacteria in the reference RNA sequences from NCBI nucleotide BLAST. The query sequences were used to create a phylogenetic tree using the neighbor-joining method, where the bootstrap test included 2000 replicates [16].

### 2.4. FT-IR for Pesticide-Degrading Activity Confirmation

The FT-IR spectroscopy was performed as described by Pourbabaee et al. (2018) [16] to obtain information about the qualitative changes of chlorpyrifos in bacteria-treated solution. Eighteen isolates were incubated in chlorpyrifos-containing liquid media for 14 days. From each culture, 5 mL was extracted using 10 mL n-hexane on a rotary shaker for 30 min, dehydrated via anhydrous sodium sulfate (Na_2_SO_4_), evaporated to dryness, subsequently diluted to a final volume of 5 mL with acetone, and analyzed by using FT-IR spectrophotometer in the range of 400–4000 cm^−1^ at 20 °C [17].

### 2.5. Biochemical Analysis

Biochemical tests were performed to characterize each isolate. Fresh bacterial isolates were grown on autoclaved nutrient agar media. The catalase and oxidase activities of the isolates were conducted as described [7]. Briefly, each isolate was incubated in Simmons Citrate agar medium, and a change of color from green to blue due to pH change indicated a positive reaction after incubation for 48 h at 37 °C [18]. The tests for Indole, Methyl red and Voges–Proskauer, urease, motility, triple sugar iron agar, glucose, maltose, lactose, and sucrose fermentation were performed as described [7,19]. The activities of cell wall hydrolytic enzymes, including cellulase, xylanase, pectinase, amylase, and protease, were performed in minimal nutrient agar media containing carboxymethylcellulose, oat-spelt xylan, pectin, starch, and casein powder (1%) as the sole source of carbon [15,20]. The lignin derivatives’ degrading activity of the isolates was confirmed using their growth on aromatic-dye-enriched minimal nutrient media [20,21]. A test tube containing the pure culture of endophytic bacteria was inoculated with 10 mL of phenol red broth supplemented with 1 g/100 mL of various sources of carbohydrates and incubated for 24 h at 37 °C. A yellow color indicated positive reaction, and the bubbles trapped inside the Durham tube indicated gas production [18]. The pure colonies were subjected on specific substrate agar plates, as previously described [15].

### 2.6. Indole-3-Acetic Acid (IAA) and ACC Deaminase Production

The quantification method for IAA of the endophytes was adopted as previously described [22]. To measure the production of IAA, bacterial isolates were inoculated into 0.5 mg L-tryptophan/mL containing medium and at 37 °C with continuous shaking at 125 rpm for 48 h, as described [23]. Then, the 2 mL culture was centrifuged at 15,000 rpm for 1 min, and a 1 mL aliquot of the supernatant was mixed with 2 mL of Salkowski’s reagent, incubated for 20 min in darkness at room temperature. The absorbance was measured on a spectrophotometer at 530 nm, and the concentration was determined using a standard curve of pure IAA, as previously described [22].

The ACC deaminase activity of the chlorpyrifos-degrading bacteria was determined according to the modified methods [24,25,26], which measure the amount of α-ketobutyrate produced upon the hydrolysis of ACC. The endophytic bacterial strains were separately grown in tryptic soy broth medium (TSB) for 18 h at 28 °C to determine the ACC deaminase activity as described in our study [21]. The cell suspension without ACC was used as a negative control, and the one with (NH_4_)_2_SO_4_ (0.2% *w*/*v*) was used as a positive control. The number of μmol of α-ketobutyrate produced by this reaction was determined by comparing the absorbance at 540 nm of a sample to a standard curve of α-ketobutyrate ranging between 10 and 200 μmol [24,26].

### 2.7. Phosphate and Nitrogen Solubilization

The N-fixation ability of the endophytic bacteria was determined by streaking isolates on the Jensen’s medium. A yellow halo zone around bacterial growth after 5–7 days incubation at 37 ± 2 °C indicated positive N-fixation activity [7]. The phosphate solubilization by isolates growing in ‘National Botanical Research Institute’s Phosphate Growth Medium’ was observed with a halo zone [7,27].

### 2.8. Anti-Bacterial Activity against Multidrug-Resistant Bacteria

Four different multidrug-resistant human pathogenic bacteria, including *S. aureus*, *E. coli*, *Klebseilla* sp., *S. epidermidis,* were collected from Dhaka Central International Medical College and Hospital, Dhaka, Bangladesh. Sensitivity tests of all endophytes against human pathogenic bacteria were conducted using the procedures described [15]. The diameters of the inhibition zones were measured in millimeters after 16, 32, and 48 h of inoculation [15]. Since the multidrug-resistant (MDR) pathogenic bacteria did not respond to any known antibiotics or known strains within our capacity, we used *Bacillus* sp. Strain HSTU-10 (MG582603) as a negative control.

### 2.9. Rice Plant Growth-Promoting Effects of Endophytic Bacteria

The crew members of each consortium were deliberately selected to incorporate a diverse range of bacterial genera, aiming to maximize the combined effects of promoting plant growth in rice plants. The following four synthetic consortia were created using various combinations of the endophytic bacterial isolates to test the effectiveness of growth-promoting activities of the pesticide-degrading endophytic bacteria: consortium-1: *Klebsiella* sp. HSTU-Bk11, *Acinetobacter* sp. HSTU-Abk29, *Citrobacter* sp. HSTU-Abk30, and *Enterobacter cloacae* HSTU-Abk39; consortium-2: *Enterobacter cloacae* HSTU-Abk37, *Enterobacter* ludwigii HSTU-Abk40, *Acinetobacter baumannii* HSTU-ABK42, *Klebsiella* sp. HSTU-Abk31, *Acinetobacter* sp. HSTU-Bk12; consortium-3: *Pseudomonas* sp. HSTU-Bk13, *Citrobacter* sp. HSTU-Bk14, *Acinetobacter* sp. HSTU-Bk15, *Acinetobacter* sp. HSTU-Abk32, and *Burkholderia* sp. HSTU-ABK33; and consortium-4: *Acinetobacter* sp. HSTU-Abk34, *Enterobacter* HSTU-Abk36, *Enterobacter* sp. HSTU-Abk38, and *Serratia marcescens* HSTU-Abk41. All endophytes were tested for their compatibility for consortium preparation [28]. In short, each endophytic isolate was grown in nutrient broth for 24 h (10^6^ CFU/mL for rice plant) and was used as inoculum. One loopful of each endophytic isolate was streaked on the opposite side of the medium in a Petri plate and then incubated at 30 °C for 48–72 h [28]. The endophytic bacterial effects on the growth promotion of rice plant genotype (*Shonamukhi*) were assayed at germination along with the vegetative-to-paddy yield stages. To this end, the germination test was performed with individual endophytes and their consortia in Petri plates. The effects of endophytes on the vegetative and reproductive stages were assessed with synthetic consortia.

### 2.10. Seed Germination Performance

Twenty-five rice seeds sterilized with 70% ethanol for 5 min and washed with distilled water five times in a septic condition were plated on a Petri dish containing sterilized 1% agar media. Autoclaved distilled water was used for all experiments. Bacterial treatment was provided individually at 100 µL/Petri dish after 24 h (approximately 10^6^) CFU/mL, except for the control plate. Following the sowing of seeds, germination was recorded at 24 h intervals and continued up to 6 days, when the seed was considered germinated, as the plumule and radicle were >2 mm long [29]. The bacterial inoculum was prepared following a previously published method [30]. The effects of both the individual members of the bacterial consortia and the effects of each of the four consortia were recorded. The germination percentages of the seeds were calculated. The root and shoot lengths of individual seedlings were measured after 7 days of sowing [29], and the vigor index was calculated using percent germination multiplied by seedling length (shoot length + root length).

### 2.11. Effect of Endophytic Consortia on Growth of Rice Plant and Yield of Grains

The pot experiments under natural conditions were conducted in the paddy field of Hajee Mohamad Daesh Science and Technology University, Dinajpur, Bangladesh, following appropriate procedures, as published by Das et al. (2022) [7]. Dried, sterilized, and pulverized soil was used to prepare eleven pots, and the experiment layouts were as follows: Control = Only Soil; Fer = Soil + Fertilizer (100% urea); Com+ = Compost + Fer; Con-1 = Soil + Consortium-1; Con-2 = Soil + Consortium-2; Con-3 = Soil + Consortium-3; Con-4 = Soil + Consortium-4; Com + Con-1 = Com + Soil + Consortium-1; Com + Con-2 = Com + Soil + Consortium-2; Com + Con-3 = Com + Soil + Consortium-3; and Com+ Con-4 = Com + Soil + Consortium-4. The bacterial treatments were performed in triplicate, and their agronomic data were recorded.

### 2.12. Chlorophyll Content of Fresh Leaf

Prior to measuring the chlorophyll content, the fresh leaves of rice plants were weighed. Using the spectrophotometric approach, the chlorophyll was extracted in 80% acetone, centrifuged, and the absorption of the extracts at wavelengths of 663 nm (D663) and 645 nm (D645) was measured. Total chlorophyll (Chl-t), chlorophyll a (Chl-a), and chlorophyll b (Chl-b) concentrations were estimated as described (Zhang et al., 2009) [31].

### 2.13. Root Length, Shoot Length, and Plant Height

The root length was measured from the collar region to the tip of the longest root in centimeters, while the shoot length was measured from the collar region to the tip of the shoot, and mean shoot length was expressed in centimeters. The plant heights were measured from the ground level to the tip of the topmost leaf at early stages (15, 30, 60, 90 days), up to the tip of the main panicle at maturity, and the average height was expressed in centimeters [32].

### 2.14. Plant Dry Matter Production

The shoots, roots, and leaves were first washed and then air-dried in the shade for 24 to 36 h prior to weighing, and the average dry weight of the plant was expressed in grams [32].

### 2.15. Yield Parameters (Grain Yield per Plant)

The weight of the grains in the panicles per plant from five tillers selected from randomly labeled plants was recorded, and the mean was expressed in grams [32].

### 2.16. Harvesting and Observations

The paddy crops were harvested after 120 days of transplantation. The germination, seedling growth parameters, plant growth parameters, plant biomass production, and yield parameters were recorded [32].

### 2.17. GC–MS/MS Analysis of Chlorpyrifos Degradation by Each of the Four Consortia

To ensure the consortia chlorpyrifos degradation compatibility, the chlorpyrifos (1% as carbon sources) enriched media were treated with each of the four synthetic consortia for 14 days. To perform the GC–MS analysis, 5 mL of consortia treated broth was shifted to separating funnels. Next, 25 mL of deionized water and 5 mL of n-hexane were added to the separating funnel as described [7]. After 5–10 min of shaking, the n-hexane layer with the solvents, which appeared on the upper hexane layer, was kept for further analysis with the Shimadzu GCMS-QP2010 Ultra (Japan) mass detector connected with a capillary column of Rxi-5ms, 30 m long, 0.25 mm i.d., 0.25 μm film thickness. One microliter of the sample was injected in a splitless mode, and the analyses were performed in a full scan mode, ranging from *m*/*z* 50 to 400. The compounds were detected after analyzing the mass spectrum of each component using the NIST11 library [33].

### 2.18. Statistical Analysis

The data were analyzed (frequency, homogeneity of variances, and LSD (ρ < 0.05)) and visualized (graph and bar chart) using SPSS, Microsoft Excel, and R language statistical software. In the bar charts, the means and error bars depict standard errors, while each letter indicates significant (ρ < 0.05) differences in plant growth parameters between treatments.

## 3. Results

### 3.1. Isolation and Selection of Chlorpyrifos-Degrading Endophytic Bacteria

Forty isolates obtained from the roots, shoots, and leaves of *Kalijeera* (indigenous variety) and BRRI-28 rice plants were screened based on their capabilities of utilizing chlorpyrifos as their sole carbon source—a characteristic of pesticide-degrading bacteria. Out of these forty, we selected eighteen isolates by analyzing morphological (size, shape, color) data and biochemical test results. As depicted in Figure 1, all strains demonstrated noticeable growth over 12 days compared to that observed in control.

### 3.2. Biochemical Characterization of the Pesticide-Degrading Endophytic Bacteria

The biochemical properties of each strain are summarized in Table 1. All eighteen strains demonstrated positive tests for oxidase, catalase, citrate utilization test, triple sugar iron (TSI), as well as lactose, sucrose, dextrose, and maltose fermentation. Furthermore, all strains except HSTU-Abk29, HSTU-Abk32, HSTU-Abk33, and HSTU-Bk14 showed positive motility. Except for strains HSTU-Abk30, HSTU-Abk31, HSTU-Bk12, HSTU-Bk15, HSTU-Abk33, and HSTU-Abk36, all strains tested positive for urease. Strains HSTU-Abk30, HSTU-Abk33, and HSTU-Bk15 were positive in the Voges–Proskauer test but negative in the methyl red (MR) test. Cell wall hydrolytic enzymes, amylase, proteases, and xylanase were secreted by all strains except strain HSTU-Bk12, which appeared to be negative for xylanase. The cellulase enzyme was secreted by all strains except HSTU-Abk34, HSTU-Bk13, HSTU-Bk14, and HSTU-Bk15. In addition, the ligninolytic enzyme secretion by all strains was assayed against several dye compounds enriched with minimal nutrient media. The assays revealed that all strains had the abilities of degrading dyes, including trypan blue, congo red, toluidine blue, avitera blue, and bromothymol blue, except strains HSTU-Abk29 and HSTU-Abk32, which were unable to degrade bromothymol blue and toluidine blue, respectively (Table 1).

### 3.3. Molecular Characterization of the Pesticide-Degrading Endophytic Bacteria

The vigorous mineralizing capabilities of chlorpyrifos (CPF) by five endophytic bacteria, including *Klebsiella*, *Enterobacter*, *Citrobacter*, *Serratia*, and *Acinetobacter* isolated from the roots of rice plants, are illustrated in Figure 2A. Two strains of *Enterobacter* sp. HSTU-Abk38 and HSTU-Abk36 showed substantial genetic distances and occupied different taxa in the phylogenetic tree (Figure 2A). In particular, the strain HSTU-Abk38 was placed in a separate node located between the *Klebsiella pneumoniae* and *Klebsiella aerogenes*. Similarly, the *Citrobacter* sp. Strain HSTU-Abk30 was deviated to a single node from the *Citrobacter freundi* and *Enterobacter ludwigi* (Figure 2A). A very similar observation was recorded for the *Serratia marcescens* strain HSTU-Abk41 and *Acinetobacter* sp. Strain HSTU-Abk29, suggesting genetic diversities of these strains.

Four genera, *Acinetobacter*, *Pseudomonas*, *Citrobacter*, and *Enterobacter,* isolated from the shoots of rice plants (Figure 2B), were not placed in the same node or sister taxa with reference type strains, except the *Acinetobacter* sp. strain HSTU-Bk15, which showed 100% similarity with the *Acinetobacter soli* strain B1. Notably, the *Citrobacter* sp. strain HSTU-Bk14 was placed in sister taxa with the *Enterobacter cloacae* strain HSTu-ABk39, which formed a different node from the *Citrobacter* and *Enterobacter* (Figure 2B). Similarly, the *Pseudomonas* sp. strain HSTU-Bk13 occupied a single node, which was closer to the *Pseudomonas aeruginosa* strain DSM50071 and *Pseudomonas aeruginosa* strain ATCC10145. Overall, these results suggest that the chlorpyrifos mineralizing endophytes isolated from the shoots exhibit diversities but are dominated by the *Enterobacter* species.

A total of three genera of endophytic bacteria, i.e., *Klebsiella*, *Burkholderia*, and *Acinetobacter,* were obtained from the leaves of rice plants (Figure 2C). Four strains were placed with the same taxon of *Klebsiella*, *Burkholderia*, and *Acinetobacter* species. The *Acinetobacter baumannii* strain HSTu-ABk42 was placed in sister taxa with the *Acinetobacter baumannii* strain ATCC 19606, while the *Acinetobacter* sp. strain HSTu-ABk32, *Acinetobacter* sp. strain HSTu-ABk34, and *Acinetobacter* sp. strain HSTU-Bk12 were separately placed in different nodes of the same clade. Furthermore, the *Burkholderia* sp. strain HSTu-ABk33 occupied a sister taxon with the *Burkholderia territorii* strain LMG28158, which greatly deviated from the other *Burkholderia* nodes (Figure 2B). In aggregate, these results demonstrate that the chlorpyrifos mineralizing leaf endophytes of rice plants were dominated by the *Acinetobacter* species.

### 3.4. Chlorpyrifos Biodegradation Confirmation Using FT-IR Analysis

Figure 3 illustrates the FT-IR spectrum of chlorpyrifos biodegradation observed after 14 days of incubation of the endophytic strains with minimal nutrient media (MSN) enriched with chlorpyrifos, when the C–H asymmetric vibration bond belonging to the typical methyl at 2870–2960 cm^−1^ and 772 cm^−1^ completely disappeared after bacterial treatment. The peak around 1370–1462 cm^−1^ indicates the C=C and C=N bonds along with the peak at 1220 cm^−1^, representing the C-N bonding vibration observed in control but one that disappeared in the case of all endophyte treatments. Moreover, the peak at 1024 cm^−1^ assigned for C=O appeared only in untreated control samples (Figure 3). Notably, some new peaks around 640–650 cm^−1^, 1100–1120 cm^−1^, and 3200–3250 cm^−1^ were recorded for the samples treated with endophytic strains. These results indicate that all endophytic strains belonging to the genera of *Klebsiella*, *Enterobacter*, *Citrobacter*, *Serratia*, *Acinetobacter*, *Pseudomonas*, and *Burkholderia* isolated from rice plants were capable of using chlorpyrifos as a carbon source.

### 3.5. Plant-Growth-Promoting Traits of the Pesticide-Degrading Endophytic Bacteria

#### 3.5.1. N-Fixation and PO_4_- Solubilization Activity

The nitrogen fixation activities of the pesticide-degrading endophytic bacteria were assayed in nitrogen-free Jensen’s growth media, as presented in Figure 4. Among the root endophytes, the strains *Klebsiella* sp. HSTU-Bk11, *Acinetobacter* sp. HSTu-ABk29, *Enterobacter* sp. HSTu-ABk36, *Serratia marcescens* HSTu-ABk41 were grown well in Jensen’s media, which was further evidenced by the creation of a holo zone of 11–15.8 mm in diameter. However, the expansion of the holo zone was limited to 5.5–7.75 mm in *Enterobacter* sp. HSTu-ABk38 and *Citrobacter* sp. HSTu-ABk30.

In the case of shoot endophytes, three strains, *Enterobacter cloacae* HSTu-ABk39, *Enterobacter cloacae* HSTu-ABk37, *Enterobacter ludwigii* HSTu-ABk40, formed holo zones ranging from 6.48 to 7.11 mm. Notably, three other shoot endophytic strains, *Pseudomonas* sp. HSTU-Bk13, *Citrobacter* sp. HSTU-Bk14, *Acinetobacter* sp. HSTU-Bk15, demonstrated a wider holo zone spanning 10.80–14.10 mm (Figure 4) compared to those formed by the root endophytes. Interestingly, the leaf endophytes showed the best levels of nitrogen fixation capacity, as evident by the maximum holo zone formation recorded as 25.15 mm for *Acinetobacter* sp. HSTu-ABk34, followed by 15.11 mm, 14.99 mm, 13.84 mm, 12.57 mm holo zone formation in *Burkholderia* sp. HSTu-ABk33, *Acinetobacter* sp. HSTU-Bk12, *Acinetobacter baumannii* HSTu-ABk42, and *Klebsiella* sp. HSTu-ABk31, respectively. Collectively, these findings suggest that the leaf endophytes showed superior levels of nitrogen fixation capacity from the atmosphere without symbiotic association with the plants (Figure 4).

Tolubilizeization activities of the endophytes were also recorded (Appendix A). All strains showed phosphate solubilization activities in PVK agar media with formation of holo zones in the range of 7–18 mm in diameter. Notably, the root endophyte strains *Enterobacter* sp. HSTu-ABk36 and *Acinetobacter* sp. HSTU-Bk15, the leaf endophyte strains *Acinetobacter* sp. HSTU-Bk12 and *Acinetobacter* sp. HSTu-ABk34, and the shoot endophyte strain *Pseudomonas* sp. HSTU-Bk13 demonstrated a high level of phosphate solubilization (Appendix A).

#### 3.5.2. IAA and ACC-Deaminase Activity

Figure 5 illustrates the varying production capacities of indole-3-acetic acid (IAA) by the endophytes isolated from the roots, shoots, and leaves of rice plants. Eight strains, including *Acinetobacter* sp. HSTU-Bk12, *Acinetobacter* sp. HSTU-Bk15, *Klebsiella* sp. HSTu-ABk31, *Enterobacter* sp. HSTu-ABk36, *Enterobacter cloacae* HSTu-ABk37, *Enterobacter* sp. HSTu-ABk38, *Enterobacter cloacae* HSTu-ABk39, and *Enterobacter ludwigi* HSTu-ABk40, produced nearly 7.5 µg/mL of IAA, which was the highest amount compared to that produced by the remaining ten other strains. Except for the strain *Serratia marcescens* HSTu-ABk41, which produced approximately 6.0 µg/mL of IAA, the remaining nine strains produced IAA below 3.0 µg/mL.

Similarly, the ACC-deaminase production varied greatly among the eighteen endophytic strains (Figure 5). The maximum activity of 0.037~0.048 µM/mg/h was recorded for the strains *Serratia marcescens* HSTU-ABk41, *Citrobacter* sp. HSTU-Bk14, *Klebsiella* sp. HSTU-Bk11, *Acinetobacter* sp. HSTU-ABk29, *Enterobacter cloacae* HSTU-ABk39, while the lowest amount of ACC-deaminase activity of 0.005 µM/mg/h was found in the strain *Citrobacter* sp. HSTU-ABk30. The remaining strains showed a moderate level of ACC-deaminase production, ranging from 0.02 to 0.035 µM/mg/h.

### 3.6. Anti-Bacterial Activity against Multidrug-Resistant Human Pathogenic Bacteria

The growth inhibition activity of the endophytic bacterial isolates against four multidrug-resistant human pathogenic bacteria was observed (Table 2; Appendix A). The results revealed that 40% of the endophytic isolates possessed anti-bacterial activities against *S. aureus* after 16 h of treatment, which rose to 55% and 65% after 32 h and 48 h, respectively. The highest inhibition zones of 41.5 ± 1.5 and 26 ± 0.6 mm were created by *Enterobacter cloacae* HSTu-ABk39 and *Acinetobacter* sp. HSTu-ABk34, respectively, against *S. aureus* after 32 h of treatment. It is noteworthy that 35% of the endophytic isolates produced inhibition zones against *E. coli* within 16 h of treatment, which increased to 55% after 32 h; however, these were less effective compared to those treatments observed against *S. aureus*. A similarly poor activity was demonstrated by these isolates when they were tested against *Klebsiella* sp. The second highest inhibition zone of 29 ± 0.9 mm was created by *Enterobacter* HSTu-ABk36 after 48 h of treatment with *S. epidermidis*. While 55% of the isolates were unable to demonstrate any activity against *S. epidermidis*, four strains, including *Acinetobacter* sp. ABk32, *Acinetobacter* sp. ABk34, and *Pseudomonas* sp. HSTU-Bk13, showed inhibitory activities against all four pathogenic strains (*S. aureus*, *E. coli*, *Klebsiella* sp., and *S. epidermidis*).

### 3.7. Rice Plant Growth-Promoting Effect

#### 3.7.1. Effects of Individual and Consortia of Endophytes on Germination and Seedling Growth

The germination of Shunamukhi genotype rice seeds was tested after the inoculation of bacterial endophytes individually and consortia of endophytes (Table 3). There was a significant difference among the treated and control samples after 8 days. The root and shoot lengths were significantly increased after 8 and 12 days of treatment of individual endophytes compared to those treated in the control (no endophytes). The LSD value of seed germination (9.55%), shoot lengths after 8 days (1.40), shoot lengths after 12 days (2.02), root lengths after 8 days (2.73), root lengths after 12 days (2.78), and the vigor index (354.61) were recorded for plants treated with individual endophytes (Table 3; Appendix A). The germination percentages were not significantly different when treated with any consortia of endophytes. However, consortium-1 and consortium-2 treatments displayed the highest shoot growth compared with those treated with either no bacterial endophytes (control) or consortium-3 and consortium-4. In addition, the most increased vigor activity was observed for consortium-1 treated samples (Table 3; Appendix A).

#### 3.7.2. Effects of Consortia on Vegetative and Reproductive Stages and Yield

The effects of all four endophytic consortia on vegetative growth, reproductive growth, and yield of rice plants are presented in Figure 6.

#### 3.7.3. Chlorophyll Content

Rice plants treated with compost mixed with consortium-2 produced the largest amount of chlorophyll a. On the other hand, the highest levels of chlorophyll b and total chlorophyll contents were found in the fertilizer treated rice plants (control). However, the highest ratio of chlorophyll a and chlorophyll b was observed in the com + consortium-2 treated rice plants (Figure 6A). Although the urea application was reduced by 70% in consortia (1–4) treated rice plants, there were no significant differences in chlorophyll ratio between the consortia treated plants and fertilizer treated (control) plants. This result indicates that the bacterial consortia amended nitrogen from the atmosphere and/or fortified the nutrients from the added compost.

#### 3.7.4. Root and Shoot Lengths

All four consortia treated rice plants demonstrated significant shoot growth compared to those of the control and fertilizer treated plants. Consortium-3 produced maximum shoot growth activity (Figure 6B) compared to those plants treated with other consortia-1, -2, and -4. When compost was added, no significant shoot elongation of rice plants was observed among all four bacterial consortia treated plants.

There was a significant difference in rice plant root length among consortia treated plants (Figure 6C). Again, consortia-3 produced maximum root growth compared with other consortia (-1, -2, and -4) treated rice plants, as among those observed for shoots. It is important to note that both compost and consortia (1–4) treated rice plants were much more greenish and disease free compared to the control, fertilizer, and compost + fertilizer treated plants (Figure 6D(a–d)).

#### 3.7.5. Yield

Consortium-2 treated rice plants produced maximum dry weight at harvest (Figure 6E). Notably, fertilizer, compost + Fertilizer, consortium-1, consortium-2, consortium-3, and consortium-4 treated rice plants showed similar yields (1000 seeds) compared with that of the control plants (Figure 6F). In contrast, the grain weights of 10 tillers/crops were significantly higher for the compost mixed with consortium-2 treated rice plants harvested from pot experiments. In addition, pots containing compost mixed with either consortium-1, consortium-2, consortium-3, or consortium-4 increased the rice yields by 17.3%, 38.6%, 18.2%, and 39.1%, respectively, compared to those pots containing only compost mixed with fertilizer (urea).

### 3.8. Roles of Consortia of Endophytic Bacteria in Chlorpyrifos Biodegradation

The consortia treated chlorpyrifos biodegradation was further evidenced by GC–MS/MS analysis (Figure 7). The control (untreated) chlorpyrifos solution had a major peak at spectrum 88, while no mentionable peak was detected in consortium-1 treated extract. Consequently, unmatching compounds were aligned with the NIST11s library. In addition, the consortium-2 treated extracts’ GC–MS spectra showed the existence of chlorpyrifos, Phorate sulfone, Chlorpyrifos-methyl, 2-Hydroxy-3,5,6-trichloropyridine, Carbofenothion sulfoxide, Oxydisulfoton, Carbonochloridic acid, Thionodemeton sulfone, dl-(2-Thienyl)-α-alanine, Chlorpyrifos Oxon, and Diethyl methanephosphonate (Appendix A). In fact, consortium-3 treatment created several new fragments, including Phorate sulfoxide, Phosphoric acid, Acetamide, and 4-Pyridinol (Table 4). Consortium-4 treatment also generated several new fragments, including Thionodemeton sulfone, Phosphorodithioic acid, and Thiophene (Appendix A). These results suggest that each consortium has different biodegradation and mineralization mechanisms of chlorpyrifos. This may be attributed to the different endophytic bacterial compositions of the consortia with varied capacities and enzymatic activities.

## 4. Discussion

Previously, organochlorine, organophosphorus, and carbamate group pesticides were widely used in agricultural fields in Bangladesh [34]. However, due to the banning of organochlorine group pesticides by the Bangladesh Environment Conservation Act 1995 [35], organophosphorus pesticides are widely used in agriculture. Chlorpyrifos (O,O-diethyl O-3,5,6-trichloro-2-pyridyl phosphorothioate) is one such widely used organophosphate pesticide employed to control a range of insects and pests in agriculture. The application of chlorpyrifos to increase crop yield poses a health risk to humans, animals, and other organisms alike. At high levels of chlorpyrifos exposure, this inherent neurotoxin can be lethal to humans [36]. To mitigate this issue, organophosphorus pesticide-degrading endophytic bacteria are much more desirable for biofertilizer components to develop a safe and sustainable agriculture practice. Endophytes exhibit cell wall hydrolytic and lignin-related dye-degrading enzyme activities, facilitating their penetration into host plants as symbionts. Additionally, their anti-microbial activities enable them to protect themselves and resist pathogenic strains during symbiosis, thus promoting healthier plants [20,21,37,38]. Furthermore, pesticide-degrading endophytes can contribute to the remediation of soil and plants by alleviating pesticide contamination [7,37,39]. These combined properties enable endophytes to fulfill their endophytic roles in host plants, fostering the development of resilient plants, which are resistant to pests and pathogens [20,40,41]. The current study demonstrated that the application of this synthetic consortium for rice cultivation can lead to healthy and high-yielding plants without the need for pesticides. This approach holds promise for promoting safe and sustainable agriculture practices.

In this paper, we report a total of eighteen endophytic bacterial strains isolated from *Kalijeera* (Field-1) and BRRI-28 (Field-2) genotypic rice plants capable of mineralizing chlorpyrifos. Employing a culture-dependent technique, these endophytic bacterial strains were grown and reproduced with chlorpyrifos serving as the only source of carbon, demonstrating a strong pesticide-degrading capability [7,8,15]. Furthermore, these strains were evaluated for their roles in the growth promotion of Shuna6+mukhi rice plant along with their inhibitory activities toward multidrug-resistant human pathogenic bacteria.

The 16S rRNA gene sequencing analysis of these strains indicated that the eighteen isolates belonged to six different species, including *Klebsiella* sp., *Acinetobacter* sp., *Pseudomonas* sp., *Citrobacter* sp., *Burkholderia* sp., and *Serratia* sp. While a wide range of chlorpyrifos mineralizing strains were found in the root samples, *Enterobacter* sp. (50%) and *Acinetobacter* sp. (50%) were found to be the most dominant among the shoot and leaf endophytic strains, respectively. Previously, several chlorpyrifos degrading bacteria, such as the *Enterobacter* strain B14 [42] and the *Klebsiella* sp. [43], were reported. Like the present study, endophytic strains of the *Enterobacter* sp. were previously isolated from the roots and grains of rice plants by Walitang et al. (2017) [44]. Other chlorpyrifos degrading endophytes, such as the *Pseudomonas aeruginosa* strain RRA, *Bacillus megaterium* strain RRB, *Sphingobacterium siyangensis* strain RSA, *Stenotrophomonas pavanii* strain RSB, and *Curtobacterium plantarum* strain RSC, were also isolated from chlorpyrifos treated rice plants grown in China [45]. All of these eighteen strains along with four different consortia included the strains reported in this paper (Figure 2A–C), demonstrating rice plant growth-promoting traits (Table 3; Figure 6), including enhanced germination rate, and increasing the root–shoot length and yield in the presence of only 30% urea application (Table 3; Figure 6D–F). In fact, a substantial amount of rice yields was obtained with consortia treatments, mimicking the yields under full-dose urea fertilizer (100%) applications. In the present study, a synthetic consortium consisting of diverse bacterial genera was utilized to enhance the growth and development of rice plants, enabling them to withstand environmental stressors, such as drought, heat, and pathogenic infections [38,39,40,41]. This endophytic consortium demonstrates the capacity to support plants in tolerating adverse conditions and resisting pathogen attacks, thus promoting their overall resilience and health. All four consortia significantly enhanced rice grain yields in pot experiments (Figure 6F) and degraded chlorpyrifos (Figure 7). In particular, the enhanced growth (4.3–6.5 log_10_ CFU/mL) in chlorpyrifos enriched with minimal nutrient media can be attributed to the abilities of these strains utilizing chlorpyrifos as their sole carbon source, as described above in Section 3.4. While consortia-2, -3, and -4 showed several degraded fragments of chlorpyrifos, consortium-3 presented fragments containing phorate sulfoxide, phosphoric acid, and acetamide, and consortium-4 showed thionodemeton sulfone, phosphorodithioic acid, and thiophene, which suggested their varied action toward chlorpyrifos in culture media (Figure 7; Table 4; Appendix A). This probably occurred due to their different crew members in consortia exerting various sets of enzyme activities and metabolic routes. The existence of common compounds, such as TCP (2-Hydroxy-3,5,6 trichloropyridine), DEMP (Diethyl methanephosphonate), ensured that the consortia members degraded the phosphodiester bonds of chlorpyrifos. Haque et al. (2018, 2020) reported that organophosphorus hydrolase enzyme OpdA, OpdC, OpdD, OpdE of *Lactobacillus* species isolated from chlorpyrifos impregnated fermented food, e.g., kimchi samples degraded chlorpyrifos ester bonds to non-toxic compounds TCP and DETP [8,16]. However, the generation of several other chlorpyrifos derivatives detected for the first time in this study was due to the efficiency of the GC–MS techniques along with the NIST11 library search, which opens a new window for chlorpyrifos detoxification by endophytic consortia.

Almost all of the endophytic strains reported in this paper showed oxidase, catalase, xylanase, amylase, protease, and cellulase activities, which are crucial for endophytic competence. Catalase activity defends reactive oxygen species and is essential for the successful survival of colonizing endophytes during oxidative bursts by plants [46]. Previously, similar activities were reported for catalase in the *Klebsiella* sp. strain PS19 [47], for protease in the *Burkholderia* sp., *Pseudomonas* sp., *Enterobacter* sp., *Pseudomonas* sp. [48,49], and for amylase in the *Pseudomonas* sp. [48]. In our study, we found that all but one endophyte, the *Acinetobacter* sp. strain HSTU-Bk12, secreted xylanase. However, several studies revealed the *Acinetobacter* sp. as a cellulase producer, and the *Pseudomonas* sp. and *Enterobacter* sp. as cellulase and xylanase producers [7,15].

The bioremediation of dye by endophytic bacteria would be non-hazardous, environmentally friendly, and cost effective. In this study, all strains except the *Acinetobacter* sp. HSTU-ABk29 and *Acinetobacter* sp. HSTU-ABk32 degraded different dyes, including TPB, CR, TDB, ATB, and BTB. The *Acinetobacter* sp. strains HSTU-ABk29 and HSTU-ABk32 were unable to degrade BTB and BTB dyes, respectively. As reported previously (Ali et al., 2009) [50], we also found that the *Pseudomonas* sp. could decolorize dye solution or simulated effluents. In addition, Tony et al. (2009) [51] reported that the consortia of microbes were capable of complete mineralization of azo dyes. However, endophytes harboring lignin-degrading activities might be presented with an opportunity to penetrate plants, forming symbionts [52]. An endophyte, which produces cellulase, protease, chitinase, and gelatinase, can hydrolyze fungal cell walls, and can inhibit the adhesion of fungal spores to plants, will have an advantage in mitigating fungal infection and be able to serve as a bio-fungicide.

The production of ammonia by endophytes is also a desirable trait for plant growth promotion, including the early establishment of seedlings, enhanced soil fertility, and increased phosphate solubilization [53]. Oteino et al. (2015) [54] observed that the inoculation with endophytes increased phosphate solubilization and significantly improved plant growth, which might be attributed to increased growth and development of rice plants treated with all four consortia. Auxin-producing endophytes *Burkoholderia vietnamiensis* promoted rice growth and yield [49]. Nitrogen-fixing bacterium *Pantoea agglomerans* [45] increased the growth of rice roots, shoots, flag leaves, and weights. Meanwhile, *Lysinibacillus sphaericus* produced ACC-deaminase and positively modulated the ethylene level in rice plants, consequently improving the number of panicles and grains per plant, straw, grain dry weight, and N and P uptake [55,56]. Moreover, additional Zinc (Zn) fortification was observed in rice treated with endophytes, such as the *Acinetobacter* sp., *Klebsiella* sp., *Enterobacter* sp., and *Bukholderia* sp. [57].

For the first time, we report that the *Enterobacter cloacae* strain HSTU-ABk39 and *Acinetobacter* sp. strain HSTU-ABk34 showed significant inhibitory activity against pathogenic *S. aureus*. Moreover, the *Enterobacter* sp. strain HSTU-ABk36 and *Citrobacter* sp. strain HSTU-ABk15 possessing growth inhibitor traits against *S. epidermidis* can serve as sources of novel antibiotics [58]. Bacterial strains, such as the *Bacillus* sp., *Lysinibacillus*, *Streptomyces*, *Streptomyces parvulus* Av-R5, also had anti-bacterial activity against *S. aureus* [59,60,61,62,63]. Endophytes are also the storehouse of several kinds of bioactive metabolites, including phenolics, alkaloids, quinones, steroids, saponins, tannins, and terpenoids, which makes them serve as potential anti-cancer, anti-malarial, anti-tuberculosis, anti-viral, anti-diabetic, anti-inflammatory, anti-arthritis, and immunosuppressive agents, as well as helping host plants become more resistant to abiotic and biotic stresses [64].

Bacterial consortia treatment of the Shunamukhi rice plant in a pot showed significant plant-growth-promoting effects in terms of root lengths, shoot lengths, yields, and chlorophyll contents compared with the control groups (Figure 6D,F). Higher cell numbers (10^11^ CFU/mL) of a growth-promoting endophytic bacterial consortium isolated from swamp soil have shown increased rice yield [65]. Previous research showed that nitrogen-fixing endophytic bacteria treatment directly influenced grain yield parameters in rice plants [66,67]. In a similar study, Yanni and Dazzo (2010) [68] reported that rice grain yield was augmented by 41% when they used endophytic *Rhizobium leguminosarum* as the inoculant, which is in support of our findings of enhanced yields when rice plants were treated with consortium-2 and consortium-4 (Figure 6F). Hence, the increased grain yield (38–39%), along with no changes in the total chlorophyll content even under the 70% reduced doses of nitrogen fertilizer, can be attributed to the cumulative effects of the plant-growth-promoting traits possessed by each of the crew members of all four consortia in this study. Several other studies reported similar findings, namely that plant-growth-promoting traits of endophytic bacteria augmented the nutrient uptake and enhanced rice yields [68,69,70] and yields of zucchini [71] and bermudagrass [72]. The endophytic strain of *Lysinibacillus sphaericus* [56] also demonstrated various aspects of plant-growth-promoting traits, such as those of the four consortia reported in the present study, which could accelerate plant growth along with fortifying the plant structure to provide resistance to phytopathogens. This finding might be indicative of the potential promises of these naturally occurring strains for sustainable rice production. Similarly, hormones such as auxin produced by endophytic consortia enhanced the root length and volume, augmented early seedling establishment, and increased nutrient intake from the soil (Figure 6C). Rice plants treated with compost mixed with any of the four consortia resulted in more tillers and longer spikelets, which collectively contributed to the increased (38–39%) grain weights (Figure 6F). The potential rice grain yield was affected by the increased rate of leaf photosynthesis, which could have an impact on dry matter production [73]. The fertilizer and consortium treatments showed a little difference in chlorophyll concentration, but the consortium treatments produced more rice. This might be attributed to other relevant factors, such as phytohormone synthesis and the furnishing of the crop structure toward immunity against biotic and abiotic stresses.

To date, there are no published reports on the effects of a consortium comprising naturally occurring endophytes on rice plants regarding the bioremediation of pesticides and urea reduction in field conditions. We assumed that the efficacy of consortia, comprising endophytic bacteria, as well as based biofertilizer harboring bioremediation properties, provided rice genotype *Shonamukhi* with resistance against both biotic and abiotic stresses in the field rather than in laboratory or greenhouse conditions (Figure 6D). In turn, the effects of these consortia comprising different species of endophytic bacteria on rice plants resulted in increased plant growth and tiller number (Figure 6D). This can be attributed to the abilities of members of the consortia to solubilize micronutrient and/or organic matter of the compost and make it available for root uptake. This was further confirmed by comparing the reduced growth and tillering performances of rice plants grown in compost mixed fertilizer only (Figure 6D). The significant growth-promoting role of pesticide-degrading rice endophytes along with their consortia and subsequent yield is a novel finding, which can be applied to manage a safe and sustainable agricultural practice. Furthermore, the reduced amount of nitrogen fertilizer (30%) along with endophytic consortia as a biofertilizer in rice cultivation may lower the demand for chemical fertilizers, such as urea, which, in turn, might have a beneficiary economic impact on farming in Grameen Bangladesh.

## 5. Conclusions

The tendency of a quick gain is increasing, resulting in uncontrolled and improper application of pesticides and fertilizers to enhance crop yield. However, the residues from excessive applications of pesticides and fertilizers emerge with their persistence in the environment and are responsible for the death of many endophytic species, leading to agricultural disaster through minimizing plant survival, growth, and subsequent yield. We isolated and identified eighteen endophytic bacterial isolates with high activities of growth-promoting auxin (IAA), ACC-deaminase, N-fixation, P-solubilization, and lignocellulolytic enzymes. In addition, we developed four consortia of these endophytic bacteria, which led to the successful growth and yield of rice at lower doses of urea (30%). The anti-bacterial along with plant-growth-promoting properties of these naturally occurring endophytic bacteria created the potential routes for their use as pesticide degraders and growth promoters in agriculture and as sources of anti-bacterial drugs in pharmaceuticals. The application of endophytic bacteria in agriculture as a microbial inoculant will reduce fertilizer utilization, as well as agricultural health hazards. These research outputs combined with bio-stimulant technologies will help us develop the inoculants of naturally occurring pesticide-degrading bacterial endophytes to substitute synthetic chemical fertilizers, creating a safe and sustainable agricultural practice, as articulated in the Sustainable Development Goals (also known as Global Goals) adopted by the United Nations in 2015.

## Figures and Tables

**Figure 1 microorganisms-11-01821-f001:**
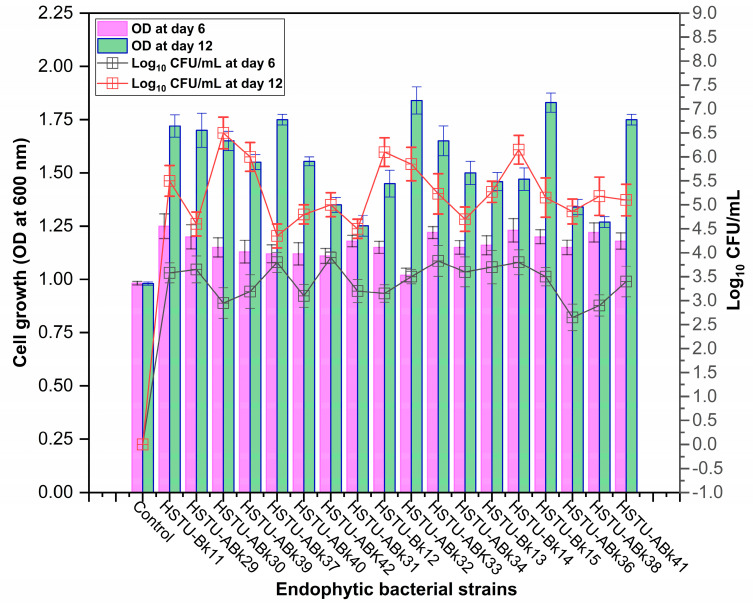
Growth performance of endophytic strains in minimal nutrient media (MSM) with chlorpyrifos (1 g/100 mL) as the carbon source.

**Figure 2 microorganisms-11-01821-f002:**
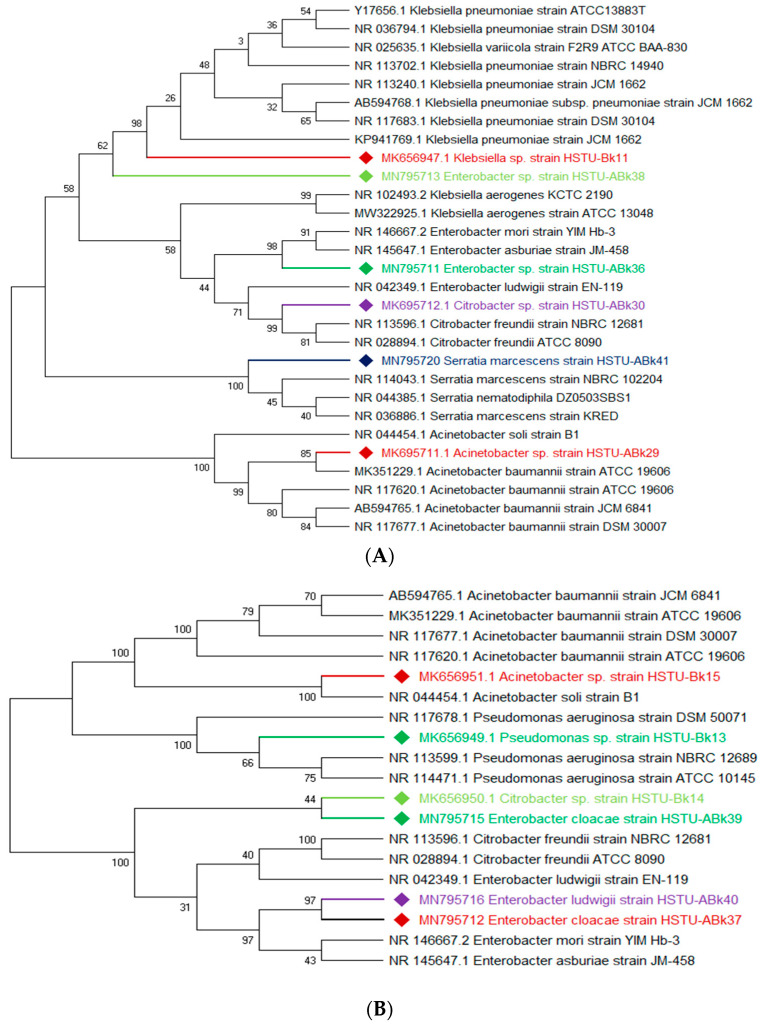
(**A**) Diversity of chlorpyrifos-resistant endophytic bacteria isolated from two different rice fields. The fields were exposed to massive pesticides for rice cultivation over the last 10 years. (**A**) Phylogeny of root endophytic bacteria. (**B**) Diversity of chlorpyrifos-resistant endophytic bacteria isolated from two different rice fields. The fields were exposed to massive pesticides for rice cultivation over the last 10 years. (**B**) Phylogeny of shoot endophytic bacteria. (**C**) Diversity of chlorpyrifos-resistant endophytic bacteria isolated from two different rice fields. The fields were exposed to massive pesticides for rice cultivation over the last 10 years. (**C**) Phylogeny of leaf endophytic bacteria. Black color strains are reference bacteria whether different colors are isolated strains in this study.

**Figure 3 microorganisms-11-01821-f003:**
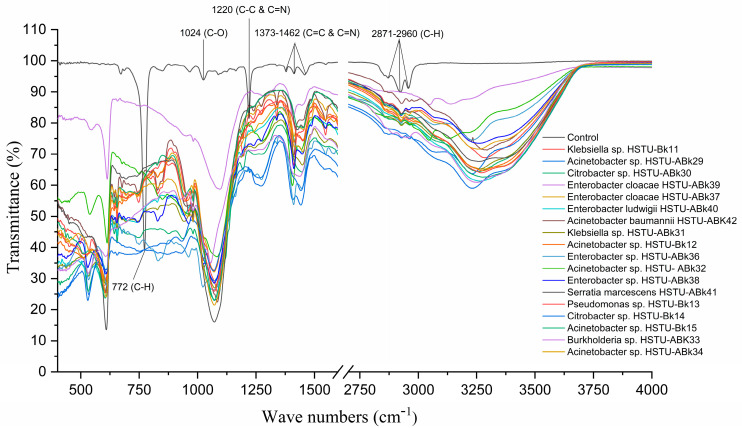
FT-IR spectra of chlorpyrifos degrading evidence of the endophytic strains in the MSM with chlorpyrifos (1 g/100 mL). The strains were grown for 14 days.

**Figure 4 microorganisms-11-01821-f004:**
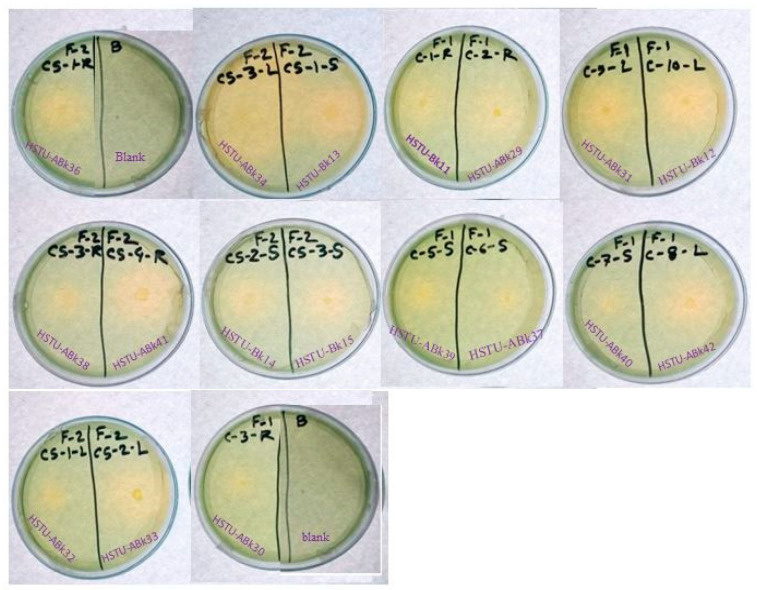
Plant-growth-promoting traits of the strains. Nitrogen fixation abilities of the endophytic strains on the nitrogen-free Jensen’s media. The diameter of the holo zone indicates the spectrum of anti-microbial activities.

**Figure 5 microorganisms-11-01821-f005:**
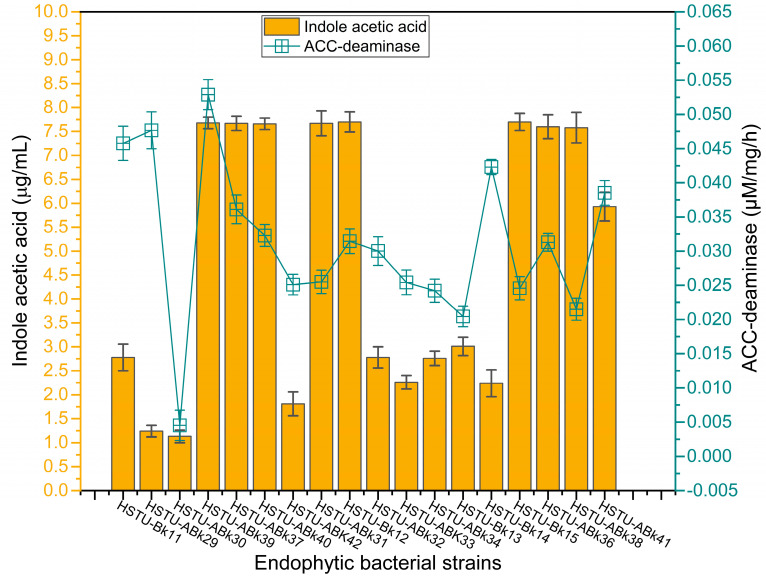
Plant-growth-promoting traits of the strains. Auxin (Indole acetic acid) and ACC-deaminase activities of the endophytic strains.

**Figure 6 microorganisms-11-01821-f006:**
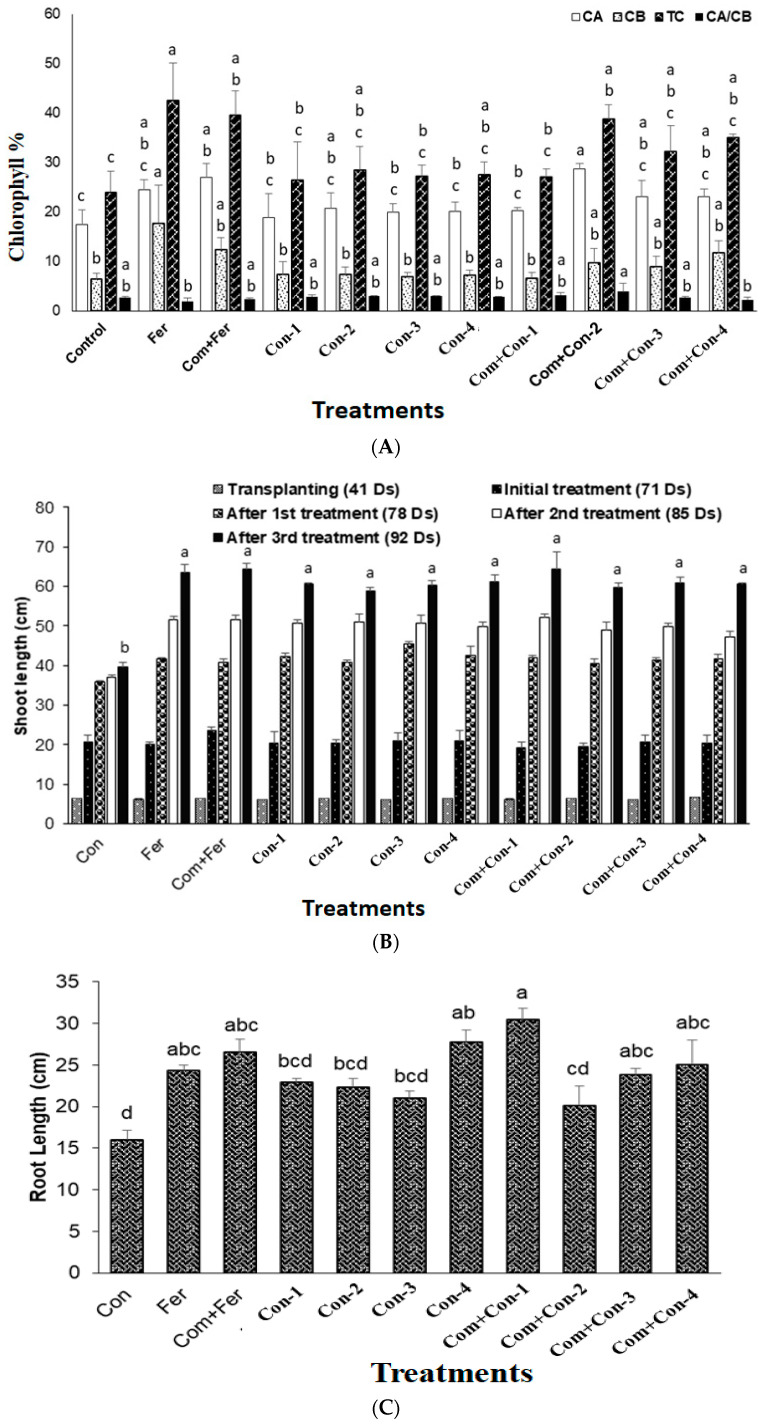
(**A**) Comparing chlorophyll content among the treated and untreated groups. (Fer, Fertilizer; Com, Compost; Bac, Bacteria; G, Bacterial consortium group). (**B**) Shoot length analysis among the endophytic bacteria in the treated and untreated groups in a time-dependent manner. (Con, Control; Fer, Fertilizer; Com, Compost; Bac, Bacteria; G, Bacterial consortium group). (**C**) Root length analysis of the endophytic bacteria. (**D**) Rice plant growth promotion with bacterial consortia at vegetative stage. (**E**) Mean of dry weights among different bacteria in the treated and untreated groups. (Con, Control; Fer, Fertilizer; Com, Compost; Bac, Bacteria; G, Bacterial consortium group). (**F**) Mean yield among different bacteria in the treated and untreated groups. (Con, Control; Fer, Fertilizer; Com, Compost; Bac, Bacteria; G, Bacterial consortium group).

**Figure 7 microorganisms-11-01821-f007:**
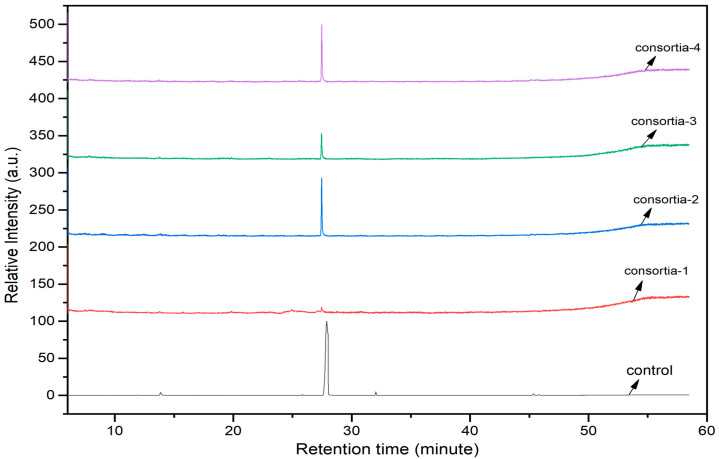
GC–MS spectra of chlorpyrifos treated with consortia-1, consortia-2, consortia-3, consortia-4.

**Table 1 microorganisms-11-01821-t001:** Summary of biochemical characterization of the endophytic bacteria isolated from rice plant genotypes Kalijera and Shonamukhi.

Isolates	Oxi	Cit	Cat	MIU	Mot	Ure	VP	MR	TSI	Lac	Suc	Dex	Mal	CMC	Xy	Amy	Pro	CR	TB	BTB	AB	TB
*Klebsiella* sp. HSTU-Bk11	+	+	+	+	+	+	+	+	+	+	+	+	+	+	+	+	+	+	+	+	+	+
*Acinetobacter* sp. HSTU-Abk29	+	+	+	-	-	+	+	-	+	+	+	+	+	+	+	+	+	+	+	-	+	+
*Citrobacter* sp. HSTU-Abk30	+	+	+	+	+	-	-	+	+ ^(Fe)^	+	+	+	+	+	+	+	+	+	+	+	+	+
*Enterobacter cloacae* HSTU-Abk39	+	+	+	+	+	+	+	-	+	+	+	+	+	+	+	+	+	+	+	+	+	+
*Enterobacter cloacae* HSTU-Abk37	+	+	+	+	+	+	+	-	+	+	+	+	+	+	+	+	+	+	+	+	+	+
*Enterobacter ludwigii* HSTU-Abk40	+	+	+	+	+	+	+	-	+	+	+	+	+	+	+	+	+	+	+	+	+	+
*Acinetobacter baumannii* HSTU-ABK42	+	+	+	+	+	+	+	-	+	+	+	+	+	+	+	+	+	+	+	+	+	+
*Klebsiella* sp. Strain HSTU-Abk31	+	+	+	+	+	-	+	+	+	+	+	+	+	+	+	+	+	+	+	+	+	+
*Acinetobacter* sp. Strain HSTU-Bk12	+	+	+	+	+	-	+	-	+	+	+	+	+	+	-	+	+	+	+	+	+	+
*Acinetobacter* sp. HSTU-Abk32	+	+	+	-	-	+	+	-	+	+	+	+	+	+	+	+	+	+	+	+	+	-
*Burkholderia* sp. HSTU-ABK33	+	+	+	-	-	-	-	-	+	+	+	+	+	+	+	+	+	+	+	+	+	+
*Acinetobacter* sp. HSTU-Abk34	+	+	+	+	+	+	+	-	+	+	+	+	+	-	+	+	+	+	+	+	+	+
*Pseudomonas* sp. HSTU-Bk13	+	+	+	+	+	+	+	+	+	+	+	+	+	-	+	+	+	+	+	+	+	+
*Citrobacter* sp. HSTU-Bk14	+	+	+	-	-	+	+	-	+ ^(Fe)^	+	+	+	+	-	+	+	+	+	+	+	+	+
*Acinetobacter* sp. HSTU-Bk15	+	+	+	+	+	-	-		+	+	+	+	+	-	+	+	+	+	+	+	+	+
*Enterobacter* sp. HSTU-Abk36	+	+	+	+	+	-	+	-	+	+	+	+	+	+	+	+	+	+	+	+	+	+
*Enterobacter* sp. HSTU-Abk38	+	+	+	+	+	+	+	-	+	+	+	+	+	+	+	+	+	+	+	+	+	+
*Serratia marcescens* HSTU-Abk41	+	+	+	+	+	+	+	-	+	+	+	+	+	+	+	+	+	+	+	+	+	+

Oxi, oxidase; Cit, citrate; Cata, catalase; MIU, motility indole and urease; Mot, motility; Ure, urease; VP, Voges–Proskauer; MR, methyl red; TSI, triple sugar iron agar; Lac, lactose; Suc, sucrose; Dex, dextrose; Mal, maltose; CMC, carboxymethyl cellulose; Xy, xylanase; Amy, Amylase; Pro, protease; Fe, iron; CR, congo red; TB, Trypan blue; BTB, bromothymol blue; AB, Avitera blue; TB, trypan blue. The symbol “+” indicates positive and “-” indicates negative.

**Table 2 microorganisms-11-01821-t002:** Minimum inhibitory concentration in mm of treated bacteria against human pathogenic bacteria.

Isolates	Multidrug-Resistant Human Pathogenic Bacteria											
	*S. aureus* (mm ± SE)			*E. coli* (mm ± SE)			*Klebshilla* (mm ± SE)			*S. epidermidis* (mm ± SE)		
	16 h	32 h	48 h	16 h	32 h	48 h	16 h	32 h	48 h	16 h	32 h	48 h
*Klebsiella* sp. HSTU-Bk11	12.75 ± 0.25	15 ± 0.2	20.5 ± 0.5	-	8.5 ± 0.5	7.5 ± 0.5	-	-	-	13.5 ± 3.5	17 ± 5.0	18.5 ± 1.5
*Acinetobacter* sp. HSTU-ABk29	-	6.5 ± 1.5	6.5 ± 1.5	-	-	-	-	-	-	9.5 ± 1.5	13 ± 2.0	12 ± 0.0
*Citrobacter* sp. HSTU-ABk30	- -	11 ± 0.0	10.5 ± 0.5	- -	- -	- -	- -	- -	- -	- -	- -	- -
*Enterobacter cloacae* HSTU-ABk39	40 ± 2	**41.5 ± 1.5 ****	40 ± 0.0	-	-	-	-	-	9 ± 1.0	-	-	9.5 ± 1.5
*Enterobacter cloacae* HSTU-ABk37	-	-	10 ± 0.0	6 ± 0.0	6 ± 0.0	6 ± 0.0	-	-	-	-	9.5 ± 1.5	10 ± 0.0
*Enterobacter ludwigii* HSTU-ABk40	-	-	-	6 ± 0.0	6 ± 0.0	6 ± 0.0	-	-	8 ± 0.0	-	-	-
*Acinetobacter baumannii* HSTU-ABK42	-	-	-	-	-	-	8.5 ± 1.5	-	-	-	-	-
*Klebsiella* sp. HSTU-ABk31	-	-	-	-	7 ± 0.0	7 ± 0.0	8.5 ± 0.5	-	-	-	-	-
*Acinetobacter* sp. HSTU-Bk12	-	-	-	-	5.5 ± 0.5	6 ± 0.0	6.5 ± 0.5	-	-	-	-	-
*Acinetobacter* sp. HSTU-ABk32	16.5 ± 0.5	16.5 ± 0.5	17 ± 0.0	8.5 ± 0.5	8.5 ± 0.5	10 ± 0.0	-	7 ± 0.0	15 ± 0.0	9 ± 0.0	10.5 ± 0.5	9.5 ± 0.5
*Burkholderia* sp. HSTU-ABK33	-	-	-	-	-	-	-	5.5 ± 0.5	5.5 ± 0.5	-	-	-
*Acinetobacter* sp. HSTU-ABk34	18 ± 2.0	**26 ± 0.6 ****	24.5 ± 4.5	12.5 ± 0.5	12.5 ± 0.5	12.5 ± 0.5	9.5 ± 1.5	9.5 ± 1.5	10.5 ± 0.5	10 ± 0.0	11 ± 0.0	10.5 ± 0.5
*Pseudomonas* sp. HSTU-Bk13	15 ± 1.0	16 ± 2.0	17.5 ± 0.5	8 ± 1.0	7.5 ± 0.5	7.5 ± 0.5		6.5 ± 0.5		18.5 ± 1.5	18.5 ± 1.5	19 ± 1.0
*Citrobacter* sp. HSTU-Bk14	10 ± 2.0	11.5 ± 1.5	11.5 ± 1.5	-	-	-	-	-	-	-	-	-
*Acinetobacter* sp. HSTU-Bk15	11.5 ± 1.5	16.5 ± 0.5	23.5 ± 1.5	-	-	-	17 ± 1.0	16 ± 2.0	16 ± 3.0	16 ± 1.0	16 ± 1.0	17.5 ± 2.5
*Enterobacter* HSTU-ABk36	-	-	-	6 ± 0.0	11.5 ± 2.5	14.5 ± 0.5	10 ± 1.0	10.5 ± 0.5	9 ± 1.0	25 ± 10.0	28.5 ± 10.5	**29 ± 9.0 ****
*Enterobacter* sp. HSTU-ABk38	-	11.5 ± 0.5	11.5 ± 0.5	-	-	-	-	6 ± 0.0	7 ± 0.0	-	-	-
*Serratia marcescens* HSTU-ABk41	-	-	-	-	-	-	-	6 ± 0.0	6.5 ± 0.5	-	-	-

The bold and ** means high activity was achieved.

**Table 3 microorganisms-11-01821-t003:** Individual and consortium effects on seedling and growth promotion.

Treatment	Germination % (Mean ± SE)	Shoot Length after 8 Days (Mean ± SE)	Shoot Length after 12 Days (Mean ± SE)	Root Length after 8 Days (Mean ± SE)	Root Length After 12 Days (Mean ± SE)	Vigor Index (Mean ± SE)
*Klebsiella* sp. HSTU-Bk11	95.56 ± 2.22 ^a^	5.00 ± 0.50 ^abcd^	6.77 ± 1.01 ^abc^	6.73 ± 1.92 ^ab^	7.43 ± 0.46 ^abc^	1116.67 ± 119.46 ^abc^
*Acinetobacter* sp. HSTU-ABk29	91.11 ± 5.88 ^ab^	5.40 ± 0.31 ^abcd^	7.37 ± 0.33 ^abc^	6.43 ± 1.32 ^abcd^	7.50 ± 0.76 ^abc^	1068.89 ± 79.08 ^abc^
*Citrobacter* sp. HSTU-ABk30	95.56 ± 2.22 ^a^	4.77 ± 0.12 ^bcd^	5.57 ± 0.58 ^bcd^	6.33 ± 1.20 ^abcd^	6.77 ± 1.65 ^abc^	1056.00 ± 83.44 ^abc^
*Enterobacter cloacae* HSTU-ABk39	88.89 ± 2.22 ^ab^	4.73 ± 0.39 ^bcd^	7.30 ± 0.80 ^abc^	3.90 ± 0.56 ^dc^	4.97 ± 0.93 ^c^	771.56 ± 105.30 ^cd^
*Enterobacter cloacae* HSTU-ABk37	93.33 ± 0.0 ^a^	5.33 ± 0.22 ^abcd^	5.50 ± 0.53 ^cd^	7.67 ± 0.33 ^ab^	6.27 ± 1.50 ^bc^	1213.33 ± 51.40 ^ab^
*Enterobacter ludwigii* HSTU-ABk40	93.33 ± 3.85 ^a^	5.33 ± 0.38 ^abcd^	7.33 ± 0.45 ^abc^	6.83 ± 0.69 ^ab^	7.83 ± 0.44 ^ab^	1137.11 ± 105.13 ^ab^
*Acinetobacter baumannii* HSTU-ABK42	88.89 ± 2.22 ^ab^	5.17 ± 0.17 ^abcd^	6.50 ± 0.29 ^abcd^	7.17 ± 0.88 ^ab^	7.50 ± 1.04 ^abc^	1098.89 ± 101.99 ^abc^
*Klebsiella* sp. HSTU-ABk31	95.56 ± 2.22 ^a^	5.90 ± 0.31 ^abc^	6.53 ± 0.30 ^abcd^	7.27 ± 0.67 ^ab^	7.07 ± 1.38 ^abc^	1256.67 ± 16.51 ^ab^
*Acinetobacter* sp. HSTU-Bk12	93.33 ± 0.00 ^a^	5.10 ± 0.10 ^abcd^	7.27 ± 0.59 ^abc^	6.07 ± 0.70 ^bcd^	6.67 ± 0.17 ^abc^	1042.22 ± 71.76 ^abc^
*Acinetobacter* sp. HSTU-ABk32	93.33 ± 0.00 ^a^	5.27 ± 0.15 ^abcd^	6.63 ± 0.93 ^abc^	6.10 ± 0.31 ^abcd^	8.03 ± 0.27 ^ab^	1060.89 ± 41.86 ^abc^
*Burkholderia* sp. HSTU-ABK33	95.56 ± 2.22 ^a^	6.27 ± 0.46 ^a^	7.00 ± 0.50 ^abc^	6.67 ± 0.41 ^abc^	9.07 ± 0.64 ^a^	1237.56 ± 87.10 ^ab^
*Acinetobacter* sp. HSTU-ABk34	88.89 ± 2.22 ^ab^	6.00 ± 0.40 ^ab^	7.57 ± 0.35 ^ab^	7.83 ± 0.60 ^ab^	8.10 ± 0.47 ^ab^	1228.89 ± 86.47 ^ab^
*Pseudomonas* sp. HSTU-Bk13	93.33 ± 3.85 ^a^	5.43 ± 0.64 ^abcd^	6.37 ± 1.10 ^abcd^	7.33 ± 0.67 ^ab^	6.70 ± 1.82 ^abc^	1196.22 ± 138.18 ^ab^
*Citrobacter* sp. HSTU-Bk14	91.11 ± 5.8 ^ab^	4.83 ± 0.33 ^bcd^	8.17 ± 0.64 ^a^	7.50 ± 0.29 ^ab^	7.33 ± 0.44 ^abc^	1128.89 ± 111.31
*Acinetobacter* sp. HSTU-Bk15	97.78 ± 2.22 ^a^	4.73 ± 0.26 ^bcd^	6.97 ± 0.75 ^abc^	5.20 ± 0.51 ^bcd^	7.20 ± 0.85 ^abc^	969.56 ± 62.34
*Enterobacter* sp. HSTU-ABk36	95.56 ± 2.22 ^a^	5.77 ± 0.64 ^abcd^	6.10 ± 0.32 ^bcd^	8.80 ± 0.76 ^a^	7.27 ± 0.93 ^abc^	1386.00 ± 100.34 ^a^
*Enterobacter* sp. HSTU-ABk38	97.78 ± 2.22 ^a^	4.40 ± 0.20 ^de^	6.53 ± 0.62 ^abcd^	6.73 ± 0.23 ^ab^	9.17 ± 0.17 ^a^	1087.11 ± 18.72 ^abc^
*Serratia marcescens* HSTU-ABk41	93.33 ± 3.85 ^a^	4.77 ± 0.37 ^bcd^	7.33 ± 0.73 ^abc^	6.00 ± 1.53 ^bcd^	7.83 ± 1.20 ^ab^	1009.78 ± 188.26 ^bc^
Control	82.22 ± 5.88 ^b^	3.27 ± 0.43 ^e^	4.60 ± 0.46 ^d^	4.00 ± 0.29 ^cd^	6.13 ± 0.52 ^bc^	600.44 ± 80.22 ^d^
LSD of individual bacteria	9.55	1.40	2.02	2.73	2.78	354.61
Consortia/Group-1	96.67 ± 3.3 ^a^	6.45 ± 0.05 ^a^	8.5 ± 1 ^a^	6.65 ± 0.35 ^a^	8.75 ± 0.75 ^a^	1267.33 ± 72 ^a^
Consortia/Group-2	93.33 ± 6.7 ^a^	6.4 ± 0.2 ^a^	8.7 ± 0.1 ^a^	5.7 ± 0.3 ^a^	8.15 ± 1.85 ^a^	1128.67 ± 71 ^ab^
Consortia/Group-3	93.33 ± 0.0 ^a^	5.5 ± 0.5 ^b^	8.8 ± 0.5 ^a^	6.5 ± 0.5 ^a^	7.15 ± 0.55 ^ab^	1050.00 ± 116 ^ab^
Consortia/Group-4	93.33 ± 00 ^a^	9.75 ± 0.25 ^ab^	7.5 ± 1 ^ab^	7.1 ± 0.1 ^a^	7.25 ± 0.25 ^ab^	1176.00 ± 37 ^ab^
Control	93.33 ± 0.0 ^a^	4.55 ± 0.05 ^b^	5.85 ± 0.35 ^b^	6 ± 1.0 ^a^	5.5 ± 0.0 ^b^	984.67 ± 98 ^b^
LSD of group treatment	8.93	1.55	2.39	1.88	2.56	273.50

Treatments with the same letter are not significantly different.

**Table 4 microorganisms-11-01821-t004:** Biodegradation profile of consortium-3 treated chlorpyrifos (1 g/100 mL) enriched with minimal broth medium.

Similarity of Hit	Search Spectrum	Soft Ionization (SI)	Spectrum	Molecular Weight (Da)	Molecular Form	Molecular Structure
1,2,3,4,5,8	76,69,67,66,65,57	2921	88	2	Chlorpyrifos	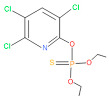
6	59	6515	38	4	2-Hydroxy-3,5,6 trichloropyridine	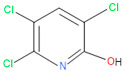
7	58	6515	38	4	2-Hydroxy-3,5,6-trichloropyridine	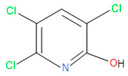
9	56	2588	3	6	Phorate sulfoxide	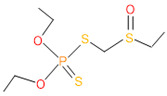
10	56	2588	3	6	Phorate sulfoxide	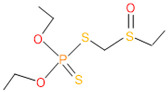
11	55	5598	13	0	Chloropyriphos-methyl	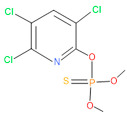
12	55	2588	3	6	Phorate sulfoxide	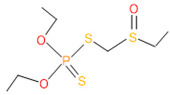
13	55	2588	4	7	Phorate sulfoxide	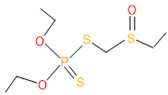
14	54	2921	88	2	Chlorpyrifos	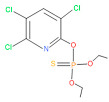
15	53	2588	4	7	Phorate sulfone	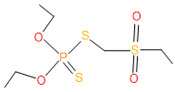
16	53	16,947	69	6	Carbonochloridic	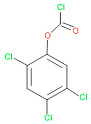
17	52	5598	15	2	Phosphoric acid	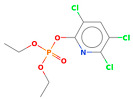
18	51	2021	58	1	dl-2-.beta.-Thienyl-.alpha.-alanine	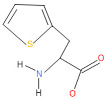
19	50	2497	7	6	Oxydisulfoton	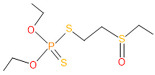
20	50	683	8	9	Diethyl methanephosphonate	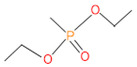
21	50	683	8	9	Diethyl methanephosphonate	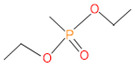
22	50	5598	13	0	Chloropyriphos-methyl	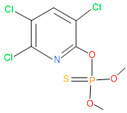
23	50	17,297	40	4	Carbofenothion sulfoxide	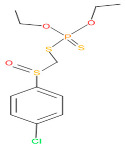
24	49	0	0	0	Acetamide,	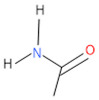
25	49	1970	40	7	4-Pyridinol,	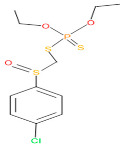

## Data Availability

The data reported in this study are contained within the article. The 16S rRNA gene sequences of the bacterial strains were deposited in NCBI and will be available immediately after acceptance of the manuscript for publication. The underlying raw data are available on request from the corresponding author.

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
