# Peer review of "Characterization of Growth-Promoting Activities of Consortia of Chlorpyrifos Mineralizing Endophytic Bacteria Naturally Harboring in Rice Plants—A Potential Bio-Stimulant to Develop a Safe and Sustainable Agriculture"

_microorganisms, 2023, doi:10.3390/microorganisms11071821_

Round 1

Reviewer 1 Report

The scientific work is interesting, exhaustive, and well-written. The topic is very interesting and actual because of the current demand to find options for soil bioremediation.

I recommend the publication of the article in Microorganisms after addressing the following issues in the manuscript:

50-52  -Please rephrase the sentence, it doesn't really make sense

122-123 – “Then, the samples were cultured into pesticide containing medium as described “- The samples were transferred again into a new pesticide containing medium after 4 days?

195 – S. aureus

434 – –S. aureus

437 – S. aureus , 440, 446  -Check the document, it should be written in small letters all over

What was used as a positive control for the analysis of antimicrobial activity? - Please add in the method section

Reviewer 2 Report

The manuscript describes the isolation of several endophytic bacterial strains from rice plants (roots, stems and leaves), different biological activities were tested, including the plat promoting profile, hydrolytic enzymes production, antibacterial activities, as well as, dye and pesticide degradation. According the multiple biological evaluations the bacterial strains with the better characteristics were those in consortium 4.

The manuscript should be accepted for publication after addressing the following commentaries

Main commentaries:

In lines 204-211, please explain, which was the rationale for the consortia conformation?

The authors carried out a great number of experiments, but need to explain how each biological activity evaluated is needed to establish the strains profiles as plant biostimulants and how contribute to assessing safe and sustainable agriculture. Plant growth-promoting activities evaluations are directly related to the title and the objective of the study, but please explain in the manuscript why it is needed to evaluate antimicrobial activity, hydrolytic enzymes production, and the dye and pesticide degrading activities? And how does the determination of these activities complement the plant growth-promoting profile of the evaluated consortia?

Additional commentaries:

In line 75, eliminate pesticides, due insecticides are pesticides

In line 82, include at least a one relevant reference to support the following information “However, commercial farming of this vital crop is under immense threat from pests, insects, and diseases along with both the biotic and abiotic stresses leading to annual loss of yield up to 50% globally”

In line 83, the fragment “54,500 metric ton pesticide was applied in 2020”, could be better “54,500 metric tons pesticide were applied in 2020”, review pertinence and adapt all similar redactions in the manuscript

In line 121 and 127, correct the centigrade symbol, be sure all centigrade symbols in the whole manuscript have the adequate format

In line 124, use adequate abbreviation for grams, g instead gm, be sure all grams abbreviation are correct in the manuscript

In line 137, use L instead l for liters, correct in whole manuscript for “µl and ml”

In line 173, add a space in “0.5mg”

In line 195, correct “S. Aureus

In lines 204-211, which was the rationale for the consortia conformation?, please explain

In line 213, correct “(106cfu/ml for Rice plant) as (106 CFU/mL for Rice plant), correct all cfu as CFU

In line 215, correct “for 48 - 72 hours” as “for 48-72 hours”

In line 282 and 285, use “p” instead “?

In figure 1, adequate the scales in the both Y axes, for better visualization of the graphic

In line 299, which means TSI?

In line 304, which means MR test?

In Table 1, correct the format of the column names, for better reading of the information

In line 321, choose and adequate synonym for “aggressive”

In line 326, Klebsiella pneumonia should be Klebsiella pneumoniae

In line 389, Figure 4A, should be Figure 4

In Figure 4A, the results of N fixation are not adequately observed, pictures quality improvement is needed

All figure 4 (A and B), conform them as just one figure with multiple panels or use continue numbering, in line 421, Figure 4B, should be Figure 5

In figure 4B, adequate the scales in the both Y axes, for better visualization of the graphic

In lines 434, 437, 440 and 446, correct “S. Aureus

In table 2, correct the bacterial species names

In table 2, review the format, values presentation is confusing

All figure 5 (A, B, C, D, E and F), conform them as just one figure with multiple panels or use continue numbering

Carefully review the manuscript and make all format and style corrections

Round 2

Reviewer 2 Report

The authors´ addressed adequately all the reviewers' commentaries on the manuscript, After corrections I consider the manuscript could be accepted for publication